# Normative Values of Height, Bodyweight and Body Mass Index of 12–17 Years Population from Extremadura (Spain)

**DOI:** 10.3390/biology10070645

**Published:** 2021-07-10

**Authors:** Rafael Gómez-Galán, Raquel Pastor-Cisneros, Jorge Carlos-Vivas, María Mendoza-Muñoz, José Carmelo Adsuar, Miguel Ángel García-Gordillo, Laura Muñoz-Bermejo

**Affiliations:** 1Social Impact and Innovation in Health (InHEALTH), University of Extremadura, 10003 Cáceres, Spain; rgomez@unex.es (R.G.-G.); jadssal@unex.es (J.C.A.); lauramunoz@unex.es (L.M.-B.); 2Promoting a Healthy Society Research Group (PHeSO), Faculty of Sport Sciences, University of Extremadura, 10003 Cáceres, Spain; jorgecv@unex.es (J.C.-V.); mamendozam@unex.es (M.M.-M.); 3Facultad de Administración y Negocios, Universidad Autónoma de Chile, Sede Talca 3467987, Chile; miguel.garcia@uautonoma.cl

**Keywords:** adolescence, anthropometric, BMI, growth table, pediatrics, youth

## Abstract

**Simple Summary:**

Growth charts constitute an essential tool for monitoring adolescents’ development. The percentile growth charts currently used are based on Basque Country population. Considering socioeconomic differences between Spanish regions, growth chart data could be inappropriate. Thus, a descriptive cross-sectional study with 4130 adolescents was conducted to describe the percentile distribution of adolescents from Extremadura and compares these percentiles with those used as reference. Bodyweight, height and BMI of Extremadura adolescents differ from the reference values currently applied. Thus, the need to use new indicators should be considered, adapted to the physical and anthropometric reality of children and young people to avoid the possible normalization of situations of thinness, overweight or obesity.

**Abstract:**

*Background*: Growth charts constitute an essential tool for monitoring adolescents’ development. In Extremadura, the percentile growth charts by Faustino Orbegozo Foundation are used. However, they are based on Basque Country population data. Considering socioeconomic differences between Spanish regions, growth chart data could not be appropriate. *Aims*: to describe the percentile distribution of adolescents from Extremadura and compare these percentiles with those proposed by the Faustino Orbegozo Eizaguirre Foundation that are currently applied in the Extremadura Health Service. *Methods:* A descriptive cross-sectional study was conducted. A total of 4130 adolescents (12–17 years) participated into the study. Bodyweight and height were assessed. *Results:* Significant differences were found comparing real measured values with commonly used reference tables for bodyweight at all ages between 12 and 13 years and at 14 years in both gender (*p* < 0.05). Differences were also found in boys at 15, 16.5 and 17 years (*p* < 0.05). Regarding height, significant differences were reported at 12, 13, 14.5, 15, 16.5 and 17 years old (*p* < 0.05) in males; while females’ results only revealed differences at 12, 12.5, 14.5 and 15.5 years (*p* < 0.05). BMI outcomes showed differences in both gender at 12, 12.5, 13, 14 and 15 years old (*p* < 0.05). Differences were also found at 16 and 14.5 years for boys and girls, respectively (*p* < 0.05). *Conclusion:* Bodyweight, height and BMI of adolescents from Extremadura differ of the reference values currently applied. Hence, this study’s outcomes suggest the need to use new indicators, adapted to the physical and anthropometric reality of children and young people to avoid the possible normalisation of situations of thinness, overweight or obesity.

## 1. Introduction

Intergenerational changes in bodyweight, height, and body mass index (BMI) have been exposed in different long-term trend studies on physical characteristics of children and adolescents [1,2]. These indices of physical development show upward trends, reflected by increasing height and body mass in successive generations of young people. This phenomenon is widespread in Europe [3].

Focusing in Spain, the 2008 data [4] show a clear long-term acceleration in population’s height and bodyweight compared to other Spanish studies that were carried out 20 years ago in individuals from Catalonia [5,6], observed Galicia [7], Madrid [8], Murcia [9], the Canary Islands [10] and Bilbao [11]. The long-term acceleration of growth is interpreted as higher values for height at all percentiles for both genders, but more accentuated in females. Regarding bodyweight values, an increase has been also observed for percentiles below or equal to the 50th percentile (BMI increase from-0.1 to 1.4) and disproportionately for higher percentiles (BMI increase from 1.5–5.3), particularly for the 97th percentile (BMI increase from 3.7 to 5.3) [4].

Growth charts constitute an essential tool for monitoring children’s and adolescents’ development and are commonly used to diagnose and control changes in height, bodyweight and BMI [12]. In Extremadura, the percentile growth charts and graphs published by the Faustino Orbegozo Eizaguirre Foundation, since its first edition in 1985, have been taken as a reference [13]. Although the Foundation updated the growth charts in 2011 through the Bilbao Growth Study: growth curves and charts (cross-sectional study) [14], currently, the health records of children in Extremadura maintain the use of tables and graphs produced and published in 2004 (FO04) [11] derived from the Growth Curves and Tables Study (longitudinal and cross-sectional studies).

It should be noted that the monitoring of child and adolescent growth in Extremadura is being carried out by means of tables based on anthropometric data from children living in the Basque Country. Considering that differences have been found in the Human Development Index (HDI) between the most developed territories in Spain, such as Basque Country, and others least developed, such as Extremadura [15], the generalization of growth chart based on data from only one region could not be appropriate since the development and growth of young people in both territories could present significant differences adolescents considering the particular living context of every region.

No studies have analysed the percentile distribution of height, bodyweight and BMI of adolescents from Extremadura. Considering that percentiles are useful to compare several population subgroups based on an average, if these populations show differences, such comparison might be biased and inappropriate.

Furthermore, it is important to consider that growth charts are based on BMI. Although the BMI has been widely used as the standard clinic tool for determining the bodyweight status in young people [16,17], the association between this parameter and other body composition variables is unclear [18,19,20]. It may be due to the BMI representing both fat and fat-free mass, being an indicator of bodyweight instead of adiposity [21]. Thus, the use of BMI as only an indicator of bodyweight status may be questioned.

Therefore, this study aims to (1) describe the percentile distribution of adolescents from Extremadura and (2) compare these percentiles with those proposed by the Faustino Orbegozo Eizaguirre Foundation that are currently applied in the Extremadura Health Service.

## 2. Materials and Methods

### 2.1. Study Design

A descriptive cross-sectional study with a 14 month cut-off period was conducted (October 2007 to December 2008). A stratified multistage sampling was used. The units of the successive stages were provinces, healthcare areas, city typology (urban/rural), educational centres (IESO), academic year (classes where the survey is carried out), gender and pupils (surveyable persons).

### 2.2. Ethics Approval

A favourable report was obtained from the Bioethics Committee of the University of Extremadura, considering that it complied with all the relevant regulations (reference code 11/2006).

### 2.3. Participants

Thirty-nine secondary schools from Extremadura were contacted to participate in this study. A total of 4130 adolescents (2128 males and 2002 females), aged between 12 and 17 years, participated into this study. Participants had to meet the following eligibility criteria to be included in this project: (1) age: 12 to 17 years; (2) registered and/or resident in the autonomous community of Extremadura; (3) authorised by parents or legal guardians; (5) acceptance of the adolescents to participate in the study.

#### Sample Size

To provide estimates with a certain degree of reliability of the survey at Autonomous Community level, a sample of 4130 individuals between 12 and 17 years of age has been selected. Based on data from the National Institute of Statistics of Spain (www.ine.es, accessed on 16 June 2021) [22], the population for this age bracket is 74,239 in Extremadura (51.3% males, 48.7% females).

Sections are made within each stratum with probability proportional to its size. Proportional allocation between strata (health areas) was used. Within each stratum, the size of each sub-stratum (rural or urban area) was used. The type of sampling was proportional in age and gender quotas, randomly selected within the educational centre of the determined populations.

### 2.4. Procedures and Measures

Data collection was carried out in their respective educational institutions by trained and standardised health personnel. The measurements were performed under standardized conditions, following the protocol established in the Data Collection Procedure Manual, developed specifically for *Childhood Obesity Surveillance Initiative* (COSI) [23]. For measuring bodyweight and height, participants were asked to remove their shoes and socks and any heavy clothes (coats, sweaters, jackets, etc.) or accessory (pockets, belts). Height was measured with a stadiometer (Tanita Tantois, Tanita Corporation, Tokyo, Japan) placed on a vertical surface with the measurement scale perpendicular to the ground. It was measured on a standing position, with shoulders balanced and arms relaxed along the body. The outcome was taken in cm, to the nearest mm. Bodyweight was evaluated using a bioimpedancemeter (Tanita MC-780 MA, Tanita Corporation, Tokyo, Japan) and was recorded in kg, up to the nearest 100 g. BMI was calculated using the formula: bodyweight (kg) divided by height squared (m^2^).

### 2.5. Statistical Analyses

All information collected was tabulated in a database specifically designed for this purpose. Statistical analyses were carried out using IBM SPSS Statistics software, version 25. Descriptive statistics were performed for all parameters, including as measures of central tendency the mean and standard deviation (SD) for the dispersion of quantitative variables and percentiles (2, 3, 10, 15, 20, 25, 50, 50, 75, 80, 85, 90, 97 and 98) as measures of position.

Normality and homogeneity of data was checked applying Kolmogorov–Smirnov and Levene’s test, respectively. Then, differences between the Extremadura data and the reference study by Faustino Orbegozo Eizaguirre (currently used by the Extremadura Health Service) were examined by applying an Independent samples T-Test adjusted by age and gender. Alpha level was set at *p* ≤ 0.05. Hedge’s *g* effect size (95% confidence interval) was calculated to determine the magnitude of between reference methods comparisons. Effect size thresholds were interpreted, as follow [24]: >0.2, small; >0.5, moderate; >0.8, large.

## 3. Results

Table 1 shows participant’s characteristics of bodyweight, height and BMI stratified by gender and age. The number of participants assessed for each group is also indicated. Results indicates that boys experience an increase in bodyweight (12.3 kg) and height (10.2 cm) between 13.5 and 15 years old, stabilizing both in posterior ages. With respect to BMI, there is a progressive increase from 12 to 17 years, obtaining the maximum value at 15 years old. Girls’ outcomes show a progressive increase in bodyweight through age increases, obtaining maximum values at 16–16.5 years. Similar results were found for height since a progressive increase until 15 years was reported, when height stabilises. BMI follows a progressive and constant increase until 16 years, where its maximum value is reached.

Table 2 and Table 3 illustrate the comparative of our bodyweight, height and BMI values and others reported in a previous cross-sectional study by Sobradillo et al. [11] for boys and girls, respectively. Between-study comparison outcomes for bodyweight shows significant differences in bodyweight between 12 and 13 years old in both gender (boys: *p* < 0.001; girls: *p* < 0.001 to 0.003) and at 14 years old (boys: *p* = 0.002; girls: *p* = 0.005). A meaningful difference was also found in boys at 15 (*p* < 0.001), 16.5 (*p* = 0.001) and 17 years (*p* < 0.001). Regarding height, significant differences were reported in boys at 12 (*p* < 0.001), 13 (*p* < 0.001), 14.5 (*p* = 0.027), 15 (*p* = 0.005), 16.5 (*p* < 0.001) and 17 (*p* < 0.001) years, while girls’ outcomes only revealed differences at 12 (*p* < 0.001), 12.5 (*p* < 0.001), 14.5 (*p* < 0.001) and 15.5 (*p* = 0.009) years. Likewise, BMI comparisons also showed differences between studies. Coincidently, there were significant differences for BMI in both gender at 12 (boys: *p* = 0.032; girls: *p* < 0.001), 12.5 (boys: *p* = 0.001; girls: *p* < 0.001), 13 (boys: *p* = 0.027; girls: *p* = 0.007), 14 (boys: *p* = 0.007; girls: *p* < 0.001) and 15 years old (boys: *p* < 0.001; girls: *p* = 0.030). Moreover, meaningful differences were found at 16 (*p* = 0.034) and 14.5 years (*p* = 0.012) for boys and girls, respectively.

Table 4, Table 5 and Table 6 displays the percentile distribution of bodyweight, height, and BMI, respectively, by gender and age.

## 4. Discussion

The main findings of the present study show significant differences in bodyweight, height and BMI growth charts between the studied population and the reference parameters applied in the Extremadura Health System [11], for numerous age and gender groups, as previously reported studies in other regions from Spain [4,11]. These studies also showed relevant differences in some age groups for bodyweight, height, and BMI. Thus, these results question the use of one or other data as state-wide references. Therefore, taking as national reference the data reported by only one community or several communities may not be representative for the development of children and adolescents from other regions of the same country. Moreover, the situations of thinness, overweight or obesity could be normalised if inappropriate or unrepresentative benchmarks are applied.

More specifically, when comparing our study with the two previous studies [4,11], results show that bodyweight progressively increases with age in boys, finding a pronounced increase at 15 and 15.5 years old in our study and the study by Sobradillo et al. [11]. However, this pronounced increase was not found by Lezcano et al. [4]. Coincidently, the maximum values for all three studies were reached around 17 years old, being lower in our study. Similarly, the results also show a progressive increase in bodyweight with age in girls. The maximum value is reached at 16 years old and stabilises thereafter, in agreement with previous studies [4,11].

Regarding height, both genders experienced a progressive increase with age, reaching a maximum value at 16 years old. Female results agree with those reported by previous studies [4,11]. In contrast, previous studies reported a considerable increase in height (more than 3 cm) between 16 and 17 years in males [4,11] that was higher than our study outcomes.

As well as bodyweight and height, BMI increased progressively with age, in line with the findings previously presented by Sobradillo et al. [11], with a noticeable increase around 15 and 15.5 years as in our outcomes. However, this increase was not found in the cross-sectional study conducted by Lezcano et al. [4], which only showed a progressive increase in BMI with age. Coincidentally, the three studies showed maximum BMI values at 17 years old. Despite the similarities found between our study and the study by Sobradillo et al. [11], the comparative results show significant greater BMI values compared to the outcomes presented by Sobradillo et al. [11] for several age groups between 12 and 15 years. However, our BMI values were quite similar to those presented by Lezcano et al. [4]. A pronounced increase in females BMI was also observed at 16 years old, reaching a maximum of 22.38 kg/m^2^, like previous studies [4,11]. Furthermore, significant differences were found between our female BMI outcomes and those reported by Sobradillo et al. [11], for most age groups between 12 and 15 years (Table 3). Contrarily, our BMI outcomes are very similar to those obtained by Lezcano et al. [4], as happens in males.

The differences and similarities found with the studies by Sobradillo et al. [11], and Lezcano et al. [4], respectively, may be due to different reasons. Firstly, the territorial representativeness of the selected sample in every study. While Sobradillo et al. [11] obtained data from children and adolescents from a single region of Spain, Lezcano et al. [4] based their results on several autonomous communities. Furthermore, the disparity in the level of physical-sport activity between the different regions of Spain could also explain these differences. It has been shown that regions with a higher level of organised physical-sports activity have a lower rate of childhood obesity [25]. Specifically, when comparing the regions of Extremadura and Basque Country, a lower percentage of children doing organised sport in Extremadura than in Basque Country was observed (70% vs. 85% in boys and 44% vs. 70% in girls). In fact, the rate of sedentarism is 37% in Extremadura compared to 28% in the Basque Country [25].

It should be known that there are socio-demographic variables, such as socio-economic level or region [26], as well as cultural variables, that must be considered because they affect the problem. These variables must be studied if we want to carry out adequate and effective preventive programmes and strategies [27]. Considering socio-demographic factors, the differences found with respect to anthropometric parameters between children and young people in the communities of the Basque Country and Extremadura could be based, among other reasons, on the economic and educational level of both regions. Data from the State System of Education Indicators of the Ministry of Education [28] indicate that the graduation rate in compulsory secondary education in the Basque Country exceeds 70%, while in Extremadura, it is below the national average. The autonomous community with the highest gross rate of Baccalaureate graduates is the Basque Country and in terms of the rate of higher education graduates the Basque Country is in second place. However, Extremadura is at the bottom of both rankings.

According to Eurostat 2008 [29], the Gross Domestic Product (GDP) of Extremadura is 18,250 million, the GDP per capita is 16,720 euros, the unemployment rate is 18.1% and the risk of poverty is 35.3%; while the Basque Country has a GDP of 66,779 million, a GDP per capita of 30,819 euros, an unemployment rate of 8.5% and a poverty risk of 9.1%.

Furthermore, these differences between regions could also be conditioned by the classification of urban (Basque Country) and rural (Extremadura) environments. In urban environments, the access to sports facilities, team-programmed activities, transport networks, etc., can directly influence adherence to a lifestyle and affect physical activity practices or the following of healthy diets [30,31]. The Quality of Life Indicator [14] established by Eurostat, which is multidimensional in nature (material living conditions, work, health, education, leisure, etc.), could explain the differences between regions in the rates of growth and overweight/obesity in Spain [32,33].

There is also evidence of a direct relationship between long-term acceleration of growth and more favourable socio-economic and health conditions (nutrition and/or absence of disease). Thus, these allow the phenotypic expression of the maximum genetic potential for growth in the child and adolescent population [34]. For example, if we take the Human Development Index of the different Spanish Autonomous Communities as an indicator of educational, economic and health levels, the Basque Country ranks as the most developed Autonomous Community in contrast to Extremadura, which is at the bottom of the list [15]. Therefore, the reference values included in the growth tables may not be applicable from one region to another, mainly due to the difference in context and environments.

In summary, the comparison of anthropometric parameters of children and young people with specific graphs from other communities that do not correspond to their own, can lead to mistakes when these children grow by percentiles higher or lower than those stipulated as maximum or minimum [35]. Moreover, it could normalise situations of anthropometric imbalances. The existence of differences between both percentile distributions could make visible the need to use new indicators, adapted to the physical and anthropometric reality of children and young people to avoid the possible normalisation of situations of thinness, overweight or obesity.

One of the strengths of this study is the representativeness of the sample, as it was 4130 children and young people from Extremadura between 12 and 17 years old, compared to the sample of the study used to elaborate the reference tables by Sobradillo et al. [11] (6443 children and young people from Basque Country aged between 0–18 years). In addition, the comparison has been made between data obtained in similar years and is therefore more reliable. If current data were used to make the comparison, the differences could be even greater probably, due to the increase in obesity levels in recent years [4,8,9,10,11]. Nevertheless, this study also presents several limitations. Considering that the main differences occur at the youngest ages, very markedly in girls, it is necessary to point out that the present study has the limitation of having focused on individuals over 12 years old. Thus, future studies focusing on younger ages than 12 years would be of great interest. Moreover, our study does not allow us to predict how the development of these children and young people will behave, since due to its cross-sectional design it is not possible to know the growth rate of participants. In addition, it should be considered that both girls and boys have not yet been completed the muscle and skeletal development at these ages. Thus, it would be interesting if future studies consider the influence of chronological and biological age on the monitored parameters.

## 5. Conclusions

The findings of this study showed that bodyweight, height and BMI of children and young people from Extremadura differ from the reference values proposed by the Faustino Orbegozo Eizaguirre Foundation that are currently used in the Extremadura Health Service. Moreover, the development-associated changes of children and young people from Extremadura compared to those from Basque Country raise the question of which values should be considered to define the levels of development and growth in the current population. Thus, we conclude that the reference values used by Extremadura Health Service may not be applicable to children and adolescents from Extremadura due to the different contexts of regions.

Hence, this study presents growth tables and graphs adapted to the physical and anthropometric reality of Extremadura adolescents. Therefore, the results presented in this study could be an important contribution to the need to use new indicators, adapted to the physical and anthropometric reality of children and young people to avoid the possible normalisation of situations of thinness, overweight or obesity.

## Figures and Tables

**Table 1 biology-10-00645-t001:** Participants’ anthropometry stratified by age and gender.

Bodyweight (kg)			Height (cm)			BMI (kg/m^2^)		
Age	*n*	Mean (SD)	Age	*n*	Mean (SD)	Age	*n*	Mean (SD)
**Boys**			**Boys**			**Boys**		
12	207	47.89 (19.55)	12	207	153.61 (8.07)	12	207	20.15 (3.43)
12.5	109	51.24 (12.23)	12.5	109	155.48 (8.28)	12.5	109	21.05 (3.95)
13	207	52.38 (12.32)	13	207	159.00 (8.34)	13	207	20.59 (3.96)
13.5	219	53.15 (11.41)	13.5	219	160.31 (8.42)	13.5	219	20.54 (3.39)
14	206	59.10 (14.23)	14	206	165.25 (8.16)	14	206	21.49 (4.30)
14.5	212	58.29 (11.97)	14.5	212	166.24 (7.88)	14.5	212	20.96 (3.34)
15	241	65.45 (13.55)	15	241	170.55 (8.02)	15	241	22.42 (3.99)
15.5	206	64.41 (13.62)	15.5	206	170.33 (7.71)	15.5	206	22.09 (3.86)
16	212	65.89 (11.60)	16	212	172.19 (6.70)	16	212	22.16 (3.30)
16.5	196	65.43 (12.32)	16.5	196	171.27 (8.01)	16.5	196	22.22 (3.36)
17	113	65.45 (11.90)	17	113	170.87 (7.64)	17	113	22.33 (3.4)
**Girls**			**Girls**			**Girls**		
12	190	48.93 (10.23)	12	190	155.34 (6.54)	12	190	20.19 (3.59)
12.5	116	50.87 (10.69)	12.5	116	156.33 (6.33)	12.5	116	20.73 (3.72)
13	189	51.47 (10.25)	13	189	157.57 (6.20)	13	189	20.67 (3.62)
13.5	205	51.98 (9.46)	13.5	205	158.90 (5.91)	13.5	205	20.57 (3.52)
14	196	54.22 (9.30)	14	196	160.47 (6.20)	14	196	21.05 (3.44)
14.5	195	54.32 (9.40)	14.5	195	160.29 (6.35)	14.5	195	21.11 (3.27)
15	213	56.85 (10.14)	15	213	161. 35 (6.71)	15	213	21.79 (3.35)
15.5	206	54.88 (7.98)	15.5	206	161.20 (5.91)	15.5	206	21.14 (3.06)
16	190	58.76 (11.24)	16	190	161.89 (6.15)	16	190	22.38 (3.82)
16.5	203	57.19 (10.54)	16.5	203	161.51 (6.43)	16.5	203	21.90 (3.76)
17	99	56.08 (8.03)	17	99	162.64 (6.38)	17	99	21.18 (2.65)

**Table 2 biology-10-00645-t002:** Between-study comparison for bodyweight, height and BMI in boys.

Boys		Extremadura	Sobradillo et al. [11]		
	N	2128 (12 to 17 years)	6443 (0 to 18 years)		
	Age (years)	Mean (SD)	Mean (SD)	*p*	Hedge’s g (95% CI)
**Bodyweight (kg)**	12	47.89 (19.55)	44.54 (9.39)	<0.001 **	0.20 (−0.05; 0.44)
	12.5	51.24 (12.23)	46.87 (7.96)	<0.001 **	0.41 (0.12; 0.70)
	13	52.38 (12.32)	49.32 (9.63)	<0.001 **	0.26 (0.00; 0.53)
	13.5	53.15 (11.41)	53.27 (10.55)	0.879	−0.01 (−0.26; 0.24)
	14	59.10 (14.23)	55.96 (10.37)	0.002 **	0.24 (−0.02; 0.49)
	14.5	58.29 (11.97)	57.95 (12.11)	0.680	0.03 (−0.26; 0.31)
	15	65.45 (13.55)	59.71 (11.65)	<0.001 **	0.44 (0.19; 0.69)
	15.5	64.41 (13.62)	65.36 (10.57)	0.322	−0.07 (−0.34; 0.19)
	16	65.89 (11.60)	64.98 (11.82)	0.251	0.08 (−0.18; 0.34)
	16.5	65.43 (12.32)	68.27 (11.83)	0.001 **	−0.23 (−0.46; −0.01)
	17	65.45 (11.90)	70.76 (11.30)	<0.001 **	−0.45 (−0.73; −0.18)
**Height (cm)**	12	153.61 (8.07)	150.10 (6.90)	<0.001 **	0.45 (0.21; 0.70)
	12.5	155.48 (8.28)	153.95 (7.42)	0.055	0.19 (−0.09; 0.48)
	13	159.00 (8.34)	156.87 (9.12)	<0.001 **	0.25 (−0.02; 0.51)
	13.5	160.31 (8.42)	160.97 (8.23)	0.248	1.11 (0.85; 1.38)
	14	165.25 (8.16)	164.13 (8.22)	0.050	0.14 (−0.12; 0.39)
	14.5	166.24 (7.88)	165.04 (8.41)	0.027 *	0.15 (−0.14; 0.44)
	15	170.55 (8.02)	168.79 (8.31)	0.005 **	0.22 (−0.03; 0.47)
	15.5	170.33 (7.71)	170.94 (6.92)	0.258	−0.08 (−0.35; 0.18)
	16	172.19 (6.70)	172.98 (6.64)	0.089	−0.12 (−0.38; 0.14)
	16.5	171.27 (8.01)	175.32 (6.54)	<0.001 **	−0.54 (−0.77; −0.31)
	17	170.87 (7.64)	176.04 (7.35)	<0.001 **	−0.69 (−0.97; −0.41)
**BMI (kg/m^2^)**	12	20.15 (3.43)	19.64 (3.19)	0.032 *	0.15 (−0.09; 0.40)
	12.5	21.05 (3.95)	19.71 (2.56)	0.001 **	0.39 (0.10; 0.68)
	13	20.59 (3.96)	19.98 (2.84)	0.027 *	0.16 (−0.10; 0.43)
	13.5	20.54 (3.39)	20.32 (3.09)	0.326	0.07 (−0.19; 0.32)
	14	21.49 (4.30)	20.67 (2.89)	0.007 **	0.21 (−0.05; 0.46)
	14.5	20.96 (3.34)	21.12 (3.38)	0.484	−0.05 (−0.33; 0.24)
	15	22.42 (3.99)	20.89 (2.89)	<0.001 **	0.41 (0.16; 0.66)
	15.5	22.09 (3.86)	22.33 (3.09)	0.378	−0.07 (−0.33; 0.20)
	16	22.16 (3.30)	21.68 (3.58)	0.034 *	0.14 (−0.12; 0.40)
	16.5	22.22 (3.36)	22.13 (3.17)	0.698	0.03 (−0.20; 0.25)
	17	22.33 (3.4)	22.83 (3.40)	0.124	−0.15 (−0.42; 0.13)

* Significant correlation at level 0.05. ** Significant correlation at level 0.01.

**Table 3 biology-10-00645-t003:** Between-study comparison for bodyweight, height and BMI in girls.

Girls		Extremadura	Sobradillo et al. [11]		
	N	2002 (12 to 17 years)	6443 (0 to 18 years)		
	Age (years)	Mean (SD)	Mean (SD)	*p*	Hedge’s g (95% CI)
**Bodyweight (kg)**	12	48.93 (10.23)	44.00 (7.31)	<0.001 **	0.52 (0.24; 0.79)
	12.5	50.87 (10.69)	44.99 (8.88)	<0.001 **	0.58 (0.27; 0.89)
	13	51.47 (10.25)	49.21 (8.45)	0.003 **	0.23 (−0.08; 0.54)
	13.5	51.98 (9.46)	52.13 (8.08)	0.823	−0.02 (−0.31; 0.28)
	14	54.22 (9.30)	52.32 (7.96)	0.005 **	0.21 (−0.09; 0.51)
	14.5	54.32 (9.40)	54.14 (8.59)	0.779	0.02 (−0.26; 0.29)
	15	56.85 (10.14)	55.29 (9.42)	0.155	0.16 (−0.12; 0.43)
	15.5	54.88 (7.98)	54.69 (7.33)	0.729	0.02 (−0.27; 0.32)
	16	58.76 (11.24)	57.84 (8.96)	0.256	0.08 (−0.26; 0.43)
	16.5	57.19 (10.54)	56.62 (6.70)	0.438	0.06 (−0.21; 0.33)
	17	56.08 (8.03)	56.35 (8.47)	0.744	−0.03 (−0.31; 0.25)
**Height (cm)**	12	155.34 (6.54)	152.25 (7.31)	<0.001 **	0.46 (0.18; 0.73)
	12.5	156.33 (6.33)	153.40 (8.88)	<0.001 **	0.40 (0.09; 0.71)
	13	157.57 (6.20)	156.74 (8.45)	0.068	0.12 (−0.19; 0.43)
	13.5	158.90 (5.91)	159.13 (8.08)	0.582	−0.04 (−0.33; 0.26)
	14	160.47 (6.20)	161.03 (7.96)	0.210	−0.08 (−0.38; 0.21)
	14.5	160.29 (6.35)	162.35 (8.59)	<0.001 **	−0.30 (−0.57; −0.02)
	15	161. 35 (6.71)	161.00 (9.42)	0.439	0.05 (−0.23; 0.32)
	15.5	161.20 (5.91)	162.28 (7.33)	0.009 **	−0.17 (−0.47; 0.12)
	16	161.89 (6.15)	161.68 (8.96)	0.656	0.03 (−0.31; 0.37)
	16.5	161.51 (6.43)	162.14 (6.70)	0.169	−0.10 (−0.37; 0.17)
	17	162.64 (6.38)	162.56 (8.47)	0.893	0.01 (−0.27; 0.29)
**BMI (kg/m^2^)**	12	20.19 (3.59)	18.91 (2.45)	<0.001 **	0.38 (0.11; 0.66)
	12.5	20.73 (3.72)	19.02 (2.87)	<0.001 **	0.49 (0.18; 0.80)
	13	20.67 (3.62)	19.96 (2.79)	0.007 **	0.20 (−0.11; 0.51)
	13.5	20.57 (3.52)	20.55 (2.70)	0.931	0.01 (−0.29; 0.30)
	14	21.05 (3.44)	20.15 (2.69)	<0.001 **	0.27 (−0.03; 0.57)
	14.5	21.11 (3.27)	20.52 (2.96)	0.012 *	0.18 (−0.09; 0.46)
	15	21.79 (3.35)	21.29 (3.16)	0.030 *	0.15 (−0.13; 0.43)
	15.5	21.14 (3.06)	20.75 (2.35)	0.065	0.13 (−0.16; 0.43)
	16	22.38 (3.82)	22.06 (2.59)	0.249	0.09 (−0.25; 0.43)
	16.5	21.90 (3.76)	21.56 (2.49)	0.196	0.10 (−0.17; 0.37)
	17	21.18 (2.65)	21.32 (2.88)	0.601	−0.05 (−0.33; 0.23)

* Significant correlation at level 0.05. ** Significant correlation at level 0.01.

**Table 4 biology-10-00645-t004:** Percentile distribution outcomes of bodyweight by gender and age.

Bodyweight (kg)														
Age (years)	*n*	P2	P3	P10	P15	P20	P25	P50	P75	P80	P85	P90	P97	P98
Boys														
12	207	31.810	32.772	35.580	36.720	37.660	39.500	46.100	55.600	56.840	58.500	62.340	70.672	76.788
12.5	109	34.060	34.600	37.300	38.850	39.700	42.700	49.000	57.900	60.000	63.450	68.400	79.920	85.440
13	207	33.112	33.968	38.960	40.500	42.240	43.800	50.500	58.600	60.500	63.000	71.060	81.020	86.408
13.5	219	35.280	35.920	40.200	41.700	43.000	43.800	52.000	60.000	63.500	64.700	69.400	75.120	75.620
14	206	36.642	37.168	43.500	45.510	47.420	49.575	57.650	65.250	69.220	72.070	81.450	93.817	99.010
14.5	212	37.630	38.739	45.000	46.300	48.580	50.275	57.150	64.525	66.120	70.610	75.000	85.688	90.090
15	241	41.024	42.952	50.340	53.030	54.240	55.750	63.700	73.600	77.300	79.110	82.680	93.018	6.448
15.5	206	42.342	43.605	48.420	50.805	53.240	55.500	61.400	72.325	75.600	79.055	85.040	95.290	96.944
16	212	46.904	48.280	53.530	55.000	56.300	58.000	64.150	71.300	73.500	77.405	80.630	94.854	99.498
16.5	196	46.270	47.237	50.740	53.910	55.680	57.375	63.850	70.650	72.840	77.890	82.090	97.427	100.000
17	113	44.140	44.626	49.280	53.430	57.080	58.150	64.500	71.000	73.100	76.800	83.400	93.740	96.152
Girls														
12	190	31.046	31.573	36.130	38.095	39.540	42.000	48.000	56.050	58.440	60.000	62.180	69.408	71.606
12.5	116	30.258	33.771	38.710	42.050	43.400	45.325	48.800	56.175	58.300	61.180	62.620	81.727	84.682
13	189	33.720	34.400	40.500	42.550	44.000	44.800	49.300	55.700	58.700	60.000	65.500	74.900	81.920
13.5	205	34.032	36.444	41.960	42.980	44.000	45.350	51.000	57.550	58.980	60.550	63.540	73.810	79.280
14	196	37.480	39.965	44.500	45.500	46.400	47.800	53.000	58.675	61.420	63.245	68.640	75.818	79.048
14.5	195	38.176	39.064	42.560	44.440	46.720	48.000	53.000	59.000	61.260	64.940	67.440	75.288	77.564
15	213	42.612	43.084	46.220	47.200	48.080	49.200	55.200	61.450	64.840	68.430	70.300	80.392	85.720
15.5	206	42.028	42.500	45.280	47.105	47.880	48.925	54.000	59.825	61.000	62.300	64.530	73.979	75.944
16	190	41.482	41.965	47.260	49.265	50.220	51.375	57.150	63.000	64.860	70.000	73.710	86.843	90.292
16.5	203	41.508	41.696	46.140	47.960	48.980	50.100	55.500	61.800	64.540	67.240	71.280	84.192	87.184
17	99	40.200	42.000	46.000	47.500	48.300	50.000	55.400	61.500	62.900	63.200	66.500	75.200	76.100

**Table 5 biology-10-00645-t005:** Percentile distribution outcomes of height by gender and age.

Height (cm)														
Age (years)	*n*	P2	P3	P10	P15	P20	P25	P50	P75	P80	P85	P90	P97	P98
Boys														
12	207	139.160	140.000	143.000	145.000	146.000	148.000	153.000	159.000	161.000	162.000	163.200	170.000	172.520
12.5	109	138.600	141.000	147.000	147.500	149.000	151.000	155.000	160.500	163.000	165.000	167.000	171.700	172.800
13	207	140.000	141.480	148.000	150.000	152.000	154.000	160.000	164.000	166.000	168.000	169.000	174.000	176.000
13.5	219	144.000	146.600	150.000	150.000	152.000	154.000	160.000	166.000	168.000	170.000	172.000	177.400	178.600
14	206	147.000	147.210	154.700	157.000	159.000	160.000	166.000	171.000	172.000	173.000	174.000	180.000	181.000
14.5	212	147.260	149.390	156.000	158.000	160.000	162.000	166.000	172.000	173.000	174.050	176.000	180.000	182.000
15	241	151.840	153.000	161.000	163.000	164.000	166.000	171.000	176.000	177.000	179.000	180.000	185.000	187.000
15.5	206	154.000	154.000	160.700	162.000	164.000	165.000	171.000	175.000	177.000	178.000	180.000	184.790	185.860
16	212	155.780	160.000	164.000	166.000	167.000	168.250	172.000	176.000	177.000	178.000	180.000	186.610	187.740
16.5	196	150.880	152.910	160.700	165.000	166.000	167.000	172.000	176.000	178.000	179.000	181.000	185.000	187.060
17	113	149.840	152.420	162.000	164.000	165.000	166.000	171.000	176.500	178.000	178.000	179.600	185.000	185.000
Girls														
12	190	141.000	142.000	146.000	149.000	150.000	150.750	156.000	160.000	161.000	162.350	163.000	167.270	168.000
12.5	116	143.000	143.510	147.000	150.000	151.000	152.250	156.000	160.000	162.000	164.000	165.300	168.000	168.000
13	189	143.000	144.700	150.000	150.500	152.000	154.000	158.000	162.000	163.000	164.000	166.000	168.600	170.000
13.5	205	147.000	147.180	152.000	153.000	154.000	155.000	159.000	163.000	163.000	164.000	166.000	171.820	172.000
14	196	148.680	150.000	152.000	154.000	155.000	156.000	160.000	165.000	166.000	167.000	168.000	173.180	177.060
14.5	195	147.260	149.880	152.600	154.000	155.000	157.000	160.000	164.000	166.000	167.000	169.000	173.000	173.240
15	213	147.280	149.420	153.000	155.000	156.000	157.000	161.000	165.000	167.000	168.000	170.000	174.580	175.000
15.5	206	150.000	150.210	154.000	155.050	157.000	157.000	161.000	165.000	166.000	167.000	168.000	173.000	174.000
16	190	150.000	150.000	155.000	156.000	157.000	158.000	161.000	165.000	167.000	168.000	169.000	176.000	178.180
16.5	203	146.160	148.000	154.000	156.000	157.000	158.000	161.000	165.000	166.000	167.000	169.600	175.880	176.920
17	99	150.000	150.000	153.000	156.000	157.000	158.000	163.000	167.000	168.000	170.000	170.000	175.000	178.000

**Table 6 biology-10-00645-t006:** Percentile distribution outcomes of BMI by gender and age.

BMI (kg/m^2^)														
Age (years)	*n*	P2	P3	P10	P15	P20	P25	P50	P75	P80	P85	P90	P97	P98
Boys														
12	207	15.137	15.351	16.290	16.875	17.187	17.483	19.377	22.290	22.952	23.590	25.347	27.708	28.294
12.5	109	15.416	15.485	16.358	17.100	17.521	18.043	20.444	23.599	24.654	25.501	26.748	29.868	30.566
13	207	14.950	15.400	16.503	17.010	17.315	17.832	19.527	22.321	23.070	24.536	25.910	30.226	32.184
13.5	219	15.177	15.715	16.974	17.207	17.773	18.146	19.562	22.648	23.193	24.844	25.781	28.257	28.728
14	206	15.978	16.132	17.218	17.627	18.060	18.448	20.347	23.022	24.325	25.293	28.372	32.857	33.781
14.5	212	15.651	16.111	17.017	17.778	18.250	18.570	20.660	22.686	23.440	24.379	25.369	29.016	29.808
15	241	16.127	16.355	18.043	18.601	18.902	19.500	21.812	24.748	25.792	26.613	27.750	31.385	32.037
15.5	206	16.626	16.792	17.891	18.269	18.533	18.952	21.285	24.846	25.584	26.595	28.190	30.213	31.838
16	212	16.909	17.137	18.408	19.318	19.611	20.112	21.672	23.515	24.046	25.037	26.141	30.598	31.166
16.5	196	16.782	17.126	18.355	19.000	19.427	19.893	21.589	23.688	24.475	25.564	27.659	30.548	31.403
17	113	17.235	17.644	18.528	18.833	19.664	20.424	21.913	23.693	24.629	25.714	26.061	31.874	33.669
Girls														
12	190	14.199	14.618	16.198	16.646	16.975	17.512	19.605	22.499	23.168	24.358	25.585	27.631	28.823
12.5	116	14.615	15.487	16.631	17.366	17.763	18.376	20.001	22.757	23.531	24.656	25.560	30.854	32.022
13	189	15.349	15.665	16.765	17.244	17.825	18.244	20.258	22.001	22.491	23.794	25.330	29.893	31.566
13.5	205	15.256	15.468	16.715	17.136	17.501	18.102	19.979	22.737	23.262	24.204	25.065	28.579	29.569
14	196	15.446	15.760	17.422	17.810	18.339	18.663	20.355	22.624	23.380	24.260	26.156	29.646	30.279
14.5	195	15.961	16.128	17.339	17.933	18.540	18.844	20.431	23.003	23.800	24.633	26.002	28.503	29.154
15	213	16.807	17.056	18.003	18.701	18.991	19.393	21.271	23.552	24.298	25.388	26.198	29.795	30.108
15.5	206	16.308	16.728	17.849	18.256	18.636	19.017	20.733	22.615	23.103	24.086	25.373	28.665	29.514
16	190	16.737	17.106	18.516	18.881	19.281	19.835	21.732	24.035	24.594	25.212	27.769	31.879	33.589
16.5	203	16.536	16.862	18.179	18.749	19.248	19.577	20.889	23.510	24.263	25.094	26.970	31.635	32.919
17	99	16.620	17.313	18.178	18.681	18.933	19.267	20.643	22.846	23.353	24.132	24.654	28.025	28.805

## Data Availability

The datasets used during the current study are available from the corresponding author on reasonable request.

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
