# Peer review of "Normative Values of Height, Bodyweight and Body Mass Index of 12–17 Years Population from Extremadura (Spain)"

_biology, 2021, doi:10.3390/biology10070645_

Round 1
Reviewer 1 Report
The only concern I have is that the data were collected more than 10 years ago, is there any particular reason why the authors not use more updated database? Can the authors demonstrate that conclusion from these data may still be applicable and is of clinical significance?
Author Response
Dear Reviewer,
Thank you for your review of our manuscript. We have carefully considered your comments and believe that the quality of the paper has improved after incorporating your suggestions. Below are our responses to your suggestions:
The only concern I have is that the data were collected more than 10 years ago, is there any particular reason why the authors not use more updated database? Can the authors demonstrate that conclusion from these data may still be applicable and is of clinical significance?
Authors’ response: Thank you for your appreciation. The prevalence of obesity and overweight in adolescents shown by the Public Health Survey in 2006 and 2017, the latest report collected, have differed minimally in their results. In 2006, the prevalence of overweight and obesity in adolescents aged 10-14 years was 19.97% and 5.04%¸ respectively. For adolescents aged 15-17, it was 16.99% and 2.22%, respectively. Regarding the last updated report, which dates to 2017, the prevalence in adolescents aged 10-14 years was 21.35% and 4.69%, respectively. In the same way, for adolescents aged 15-17 years, the prevalence of overweight is 15.43% and obesity is 3.24 %. Therefore, considering that the prevalence of overweight and obesity is similar in both reports, the findings of this study would be applicable to the current context. Moreover, as already indicated in the manuscript, both the data shown by the Faustino Obergozo Foundation and the percentiles used are from nearby dates.

Reviewer 2 Report
The use of adolescent weight assessment using BMI is still debated with differing results. On the other hand, there is essentially agreement on the use of developmental percentile graphs to assess somatic parameters. The study is current and I see its main benefit in a clearly defined regional influence on the reference values, which is essential for the interpretation of the measured data. My only comment on the work is that it would be appropriate to discuss in the discussion the influence of chronological and biological age on the monitored parameters. It should also be noted that in both girls and boys, muscle and skeletal development has not yet been completed.
Author Response
Dear Reviewer,
Thank you for your review of our manuscript. We have carefully considered your comments and believe that the quality of the paper has improved after incorporating your suggestions. Below are our responses to your suggestions:
El uso de la evaluación del peso de los adolescentes mediante el IMC todavía se debate con resultados diferentes. Por otro lado, existe un acuerdo fundamental sobre el uso de gráficos de percentiles de desarrollo para evaluar los parámetros somáticos. El estudio está actualizado y veo su principal beneficio en una influencia regional claramente definida en los valores de referencia, que es esencial para la interpretación de los datos medidos. Mi único comentario sobre el trabajo es que sería apropiado discutir en la discusión la influencia de la edad cronológica y biológica en los parámetros monitoreados. También debe tenerse en cuenta que tanto en niñas como en niños, el desarrollo muscular y esquelético aún no se ha completado.
Respuesta de los autores: Gracias por su contribución. En nuestro trabajo, no hemos estudiado la influencia de la edad cronológica y biológica sobre los parámetros monitoreados o el desarrollo muscular y esquelético. Así, se procede a incluir ambos temas como limitaciones del estudio en el apartado correspondiente. Espero que sea suficiente.
